# Experimental Study of Mechanical Wave Propagation in Solidifying Cement-Based Composites

**DOI:** 10.3390/ma17235971

**Published:** 2024-12-06

**Authors:** Luboš Jakubka, Libor Topolář, Anna Nekorancová, Richard Dvořák, Kristýna Hrabová, Felix Černý, Szymon Skibicki, Luboš Pazdera

**Affiliations:** 1Faculty of Civil Engineering, Brno University of Technology, Veveří 331/95, 602 00 Brno, Czech Republic; 2Faculty of Civil and Environmental Engineering, West Pomeranian University of Technology in Szczecin, al. Piastów 17, 70-310 Szczecin, Poland

**Keywords:** non-destructive methods, pass-through pulsed ultrasonic method, mechanical waves, composite materials, solidification and hardening, cement

## Abstract

In this paper, a new measurement procedure is presented as an experimental study. In this experimental study, a measurement system using the pass-through pulsed ultrasonic method was used. The pilot application of the measurement setup was to monitor mechanical wave changes during the solidification and hardening of fine-grained cement-based composites. The fine-grained composites had different water–cement ratios. The measured results show apparent differences in the recorded mechanical wave parameters. Significant differences were observed in the waveforms of the amplitude increase in the passing mechanical waves. At the same time, the frequency spectra of the five most dominant frequencies are presented, where the frequency lines are clear, indicating the quality of the hydration process. Based on the results, it can be concluded that the new method is usable for fine-grained cement-based materials but is not limited to that. The advantages of this method are its high variability and non-destructive character. The experimental study also outlines the possible future applications of the pulsed passage ultrasonic method.

## 1. Introduction

New possibilities are still being sought to determine and improve individual building materials’ mechanical and deformation properties. The critical process for selecting most of the characteristics of cementitious materials is the hydration of the cement. The methods of early setting, and later hardening and maturation are being monitored. These processes are very complex and important, and, ultimately, determine the overall properties of the resulting composite. Understanding the cement hydration process is essential for solving engineering problems for specific applications regarding durability and sustainability [1].

Building materials or structures may exhibit various defects in their life cycle. The correct and early identification of these defects is very important to ensure mechanical durability, stability, and safety. Early in their life cycle, the early remediation and minimisation of future damage is possible. Proper damage detection is key to evaluating the probability of failure in various situations. The instrumentation and the test procedures used in the testing field are developing significantly [2,3,4].

This experimental study uses the pass-through pulsed ultrasonic method to describe the behaviour of cement-based composite materials in the early solidification and later hardening phases. In particular, the study aimed to compose a custom experimental setup for measuring the pulsed transient ultrasonic method.

In the pilot experiments, the evolution of the amplitude and frequency spectrum of the signals was monitored throughout the measurements. The results of this study are expected to extend acoustic methods’ capabilities and support efforts for the broader use of measurement methods in the field of non-destructive testing in the construction industry during the early solidification phase of composite materials. A sub-objective was also to expand the possibilities of using the existing instrumentation to determine the early solidification time of composite materials non-invasively.

The setting and hardening process (e.g., in cementitious composites) depends on internal (endogenous) and external (exogenous) factors. The general internal factor is the composition of the formulation (e.g., the type and number of fillers and binders, water coefficient, etc.). External factors include climatic conditions and processing technology. Because the influences above are directly reflected in the characteristics of the fresh material such as compaction, deposition, rheology, and workability (it is known that the setting times of binders are different from the setting and hardening times of concrete or mortar [5], but also the cured material such as shrinkage, strength, and elasticity, other commonly used methods or measurements (e.g., internal temperature measurements) will be used to describe the properties of composites. In this experimental study, the authors have piloted simpler systems, represented by cement pastes, to facilitate identifying the processes and the subsequent interpretation of the information obtained. The team plans to open up the possibilities for further investigations on, for example, coarse-grained composites.

Acoustic methods are classified as non-destructive testing methods. In principle, they are based on monitoring the propagation of mechanical waves in the material or structure under test. According to the arrangement of the measuring procedure, these methods can be divided into the following:Active—vibration excitation and sensing;Passive—sensing vibrations generated in the material or structure.

NDT methods, when used, will not destroy or damage the material or structure being tested. Moreover, there are no permanent changes during this testing [6,7].

The main objective of NDT is to evaluate and identify various types of (surface or internal) changes, as well as defects in the material under test that could have a negative effect on the function of a building structure, engineering or electrical product, etc. Furthermore, NDT identifies the nature and dimensions of non-corrosion defects. Since it is important to prevent product defects and to co-create conditions for eliminating the causes of these defects, this type of testing is used in testing materials both at the pre-production stage, during production, and in operation [7].

Waves, in general, can be characterised as a physical phenomenon. The essence of this phenomenon is the propagation of a specific disturbance through the substance environment, whereby disturbance means a local change in some quantity characterising the state of the substance forming the given environment (e.g., its density and pressure). The source of the wave is considered to be the body (event) which causes the disturbance (wave) and from which the disturbance subsequently propagates. Waves can be divided into electromagnetic, mechanical, and matter waves (de Broglie waves) [8,9,10].

Newton’s laws govern mechanical waves, such as waves on the water surface, sound waves, and seismic waves. This type of wave can only exist in a particular material medium, such as water, air, or a solid, composed of particles (bulk elements of the medium) between which there is a bond.

Mechanical waves are a process in which the forced oscillation of one particle in the environment is gradually transferred to other particles. The particles of the environment do not move in space but oscillate only around the equilibrium positions and simultaneously deform. The oscillation proceeds at a certain speed from its source at a series of points in the plane or space. An oscillation in which there is no transmission of the substance (i.e., no displacement in a particular direction) during the propagation of the elastic wave is called linear. This oscillation occurs when the deflections of the vibrations of the particles of the medium are sufficiently small [9,10,11].

The ultrasound method is often used for the non-invasive determination of the material properties and structure in various industries, including construction. This method uses high-frequency sound waves (above 20 kHz) that are transmitted into the material, and then their reflection and transmission are measured. From these data, material properties such as density, elasticity, strength, and porosity can be determined.

In industrial construction, an ultrasound is mainly used to test concrete and other building materials to determine their properties and integrity, such as cracks, voids, and defects. This method also investigates structures such as bridges, tunnels, and buildings where testing can be performed externally without disturbing the building elements [12].

For testing the properties of materials (using ultrasound wave propagation), an ultrasound apparatus with two probes (transmitting and receiving) is often used, which measure with sufficient accuracy the time it takes for the transmitted pulse (the excited longitudinal wavefront) to pass to the receiving probe through the measuring base [13].

## 2. Methods and Materials

The pass-through pulsed ultrasonic method is currently only an experimental method in this modification for use during the solidification of building composites. The measurement procedure developed for this method uses the items and equipment owned by the Brno University of Technology, Faculty of Civil Engineering, Department of Physics. This method determines the onset of the solidification time from the amplitude and frequency spectrum change of the ultrasonic signal passing through the solidifying material. It is, therefore, a non-invasive way of testing composite materials. In materials testing, there is currently a device called Vicasonic (Schleibinger Testing Systems, Buchbach, Germany), which determines the onset of solidification based on the change in the velocity of ultrasonic wave propagation using time measurements. However, a composite measurement system could go beyond simply determining the onset of solidification. It should be more reflective of the changes in structure during the solidification process of building materials.

The principle of the pass-through pulsed ultrasonic method is the passage of mechanical waves through the material during hydration. When a fresh mixture solidifies or hardens, the properties of the signal passing between two piezoelectric transducers change, where the first transducer acts as an exciter and the second as a receiver of the signal. The circuit diagram of the pass-through pulsed ultrasonic method is shown in Figure 1.

After the measurement, it is possible to analyse the acquired signals and determine, for example, the following:-Signal amplitude;-The change in the frequency spectrum;-The attenuation of individual signals;-After modification of the measurement setup, the ultrasonic signal’s transit time—this modification was not solved in this experimental study because it would have been substituted by commercially available equipment.

All of the above data can then be correlated with, for example, the measurement of the specimens’ solidification temperature. There are other standardised procedures, but these are usually invasive test methods. For example, needle penetration depth measurements are used for testing binders, and, in concrete or mortar testing, it is usually roller penetration resistance. In these procedures, the results of determining the onset of setting and hardening are significantly affected by the frequency of individual punctures [14,15,16].

### 2.1. Measurement of Temperature

The sensors detect the magnitude of the measured physical quantity and convert it into another (usually electrical) quantity. The electrical quantities thus obtained are then processed in further evaluation circuits.

NTC thermistors were used in this experimental study. NTC thermistors have a negative temperature coefficient of resistance, corresponding to the abovementioned phenomenon. As the temperature increases, the concentration of charge carriers increases, and the electrical resistance decreases. The thermistor is made of polycrystalline ceramic and is made of a mixture of oxides. First, a powder is made from various metal oxides (e.g., Fe_2_O_3_, TiO_2_, CuO, MnO, NiO, CoO, BaO, etc.) and stabilisers and mixed with a plastic binder. This mass is then pressed into the shape of a disc, wafer, etc. [16,17,18].

### 2.2. Materials

Composite materials (composites) are multiphase materials with at least two phases (components) and different properties. Their composition produces a material with properties that neither of the phases used alone had. In contrast, all materials retain their identity within the composite, although they interact with their surroundings. There is no merging or complete dissolution, and each phase can be physically identified. A composite system is formed by mixing and combining the components in various ways and means [19,20,21,22].

The properties of concrete are primarily influenced by its essential components’ composition and mixing ratio. Unique properties of concrete can be achieved by adding appropriate admixtures to positively impact its initial properties (workability and setting time) and final properties (strength, modulus of elasticity, resistance, and durability).

Secondary effects of the surrounding environment (e.g., humidity, temperature, and aggressive agents) influence the properties of concrete.

In practical applications, complete hydration of the cement does not usually occur because large cement grains of over 50 μm do not hydrate to the core due to the uneven dispersion of the mortar water, which is lost by premature evaporation. Therefore, treatment water plays an essential role in the cement hydration process and the development of concrete strength [23,24,25,26,27]. Cement hydration as a function of the water–cement ratio is shown in [21].

From a certain degree of filling gaps between aggregates and the minimum w/c, the highest value of composite strength is reached, and even the subsequent increase in cement dosage does not significantly reduce this strength [22].

Cement pastes with the following water coefficients were chosen as the base set of measurements: 0.33, 0.40, 0.45, 0.50, 0.55, and 0.60. The aim was to investigate the behaviour of the generated mechanical waves that passed through these solidifying pastes. The parameters studied were the amplitude and dominant frequency change for pulsed ultrasonic mechanical waves and the temperature change. Water was also added to the basic set of measurements, mainly to verify the watertightness of the assembled measuring device. Portland cement CEM I 42,5 R from the Mokrá cement plant (HeidelbergCement Mokrá-Horákov, Czech Republic) and water were chosen for the production of the cement pastes.

### 2.3. Measuring Chain

The first phase of the experimental study focused on compiling the measurement chain. In this first phase, the measurements are calibrated to ensure their repeatability, and pilot measurements of fine-grained cement-based composite materials are taken.

An essential part of this phase of the solution was the selection of a suitable exciter and sensor and their proper protection from damage, mainly due to moisture. Pilot measurements of the fine-grained composites followed the correct instrumentation to fine-tune the measurement setup and set the measurement parameters so that the instruments would be relevant and the same for testing all fine-grained composites. Furthermore, emphasis was placed on selecting measurement parameters so that subsequent measurement results could be evaluated. The individual parts of the measurement chain were assembled based on a block diagram, shown in general form in Figure 1.

A Keysight Technologies Agilent 33220A functional generator was used as a signal source to generate mechanical waves, operating in the frequency range of 1 μHz to 20 MHz (Agilent, Santa Clara, CA, USA, 2023), which was sufficient for the purposes specified. This function/arbitrary waveform generator uses a direct digital synthesis technique to produce a stable, accurate output signal for pure sine waveforms with low distortion. It also enables rectangular waveforms with fast rise and fall times up to 20 MHz and linear rise times up to 200 kHz. The amplitude range can be set from 10 mV_pp_ to 10 V_pp_ [28].

The following parameters were set for research and measurement purposes:-PULSE function with a repetition period of 10 s;-Amplitude 10 V_pp_;-Pulse width 100 μs;-Rise time 5 ns.

To sufficiently and additionally amplify the signal to the required level suitable for the exciter, a custom-made controlled power amplifier [29,30] from 3S SEDLAK was used. Power amplifiers are also called terminal amplifiers or terminal stages because they form the last stage of the amplification chain.

For this study, two custom piezoelectric sensors without integrated preamplifiers were fabricated. Sensors without integrated preamplifiers can also be used as exciters, which was the case here. One sensor was used as an exciter of the generated signal (mechanical wave), and the other as a sensor to capture the transmitted mechanical wave. These changes in the mechanical wave occur when it passes through a fine-grained composite material (e.g., cement-based) during solidification and hardening due to changes in the material’s structure. Both piezoelectric elements (exciter and sensor) were mounted in a polystyrene plate, mainly for thermal insulation reasons. Before each measurement, they were treated with petroleum jelly and coated with food film for better contact with the material and easier removal at the end of each measurement. The piezoelectric sensor, which was placed on top of the measured specimen, was loaded after potting and mounting to prevent loss of contact between the sensor and the material under test (eliminating the effect of shrinkage due to solidification and hardening of the material). The polystyrene inserts contained grooves for the probes and lead wires, which allowed precise seating on the mould for the test specimens. The mould for the composite under test was plastic and cylindrical with an outer diameter of (75.0 ± 0.3) mm (see Figure 2) and is similar in volume to a Vicat ring [31].

Figure 2 shows the specimen measurements during the test. The white bottom part is a polystyrene plate in which the receiver is placed. The receiver was embedded in a polystyrene plate so that the sensing surface was flushed with the surface of the plate. A groove was also present in the polystyrene plate of the cylindrical plastic mould. This recess prevents the mass from spilling out of the mould. The upper polystyrene plate with an exciter was treated in the same manner. The top plate is weighted such that, particularly in the initial solidification phase, the exciter is pressed against the material. The load consisted of one previously moulded specimen (created during the pretests), which was the same during all measurements.

The actual recording of the signal takes place after the mould is poured, and the sensor is mounted and loaded on a DAKEL XEDO acoustic measurement and diagnostic analyser (system) using three channels. ZD Rpety-Dakel from Prague, Czech Republic developed the measuring analysers. The main building blocks of the DAKEL XEDO system are the measuring and communication units (cards), which form the measuring instruments (boxes) in groups [32]. Within the box, communication between the units occurs over a universal high-speed bus, providing all connected units with precise time information (within 1 µs). Each box shall contain at least one communication unit through which the measurement data are transmitted to the computer and the measuring unit.

An analyser from the Institute of Physics containing a communication unit called XEDO-AE was used for the measurements. This communication unit allows connection to a PC via a standard Ethernet interface. The analysers are also equipped with signal evaluation units. Each unit of the XEDO system contains a powerful 16-bit digital signal processor (DSP). The control program for the processor is fixed in ROM and located directly on the unit’s circuit board. Although the ROM is part of the hardware part of the system, the program is considered part of the software. The PC control program allows the configuration of parameters and measurements with the connected system [32].

### 2.4. Preparation and Course of the Measurement

Before the start of the measurements, it was necessary to prepare the temperature sensors, i.e., to power them to the supply cables and to insulate these “live” connections using a protective varnish for the fitted printed connections from ELCHEMCO (Zruč nad Sázavou, Czech Republic) [33]. While the protective varnish was drying, a mould was prepared for the material under investigation. The open base of the mould was coated with shrink wrap, and the ends were fixed to make the mould watertight, and no water leaked out during the solidification of the material. A de-moulding agent was applied to the inner walls of the mould to facilitate the removal of the solid specimen from the mould later.

The receiving piezoelectric sensor was then mounted in a polystyrene template in the prepared mould. This template forms the base of the cylindrical mould. A thin layer of de-moulding agent was applied to the sensing surface, and then the mould and the polystyrene base were seated on top of this. All air pockets were manually removed during the settling process. Air pockets could affect the received signal. Air pockets can form between the sensing surface and the foil at the bottom of the mould (see Figure 2—left). A thin layer of de-moulding agent was applied to the sensing surface and then covered with foil to again prevent water from reaching the piezoelectric element. The de-moulding agent in a thin layer allows for better acoustic coupling. A prepared temperature sensor was also attached to the polystyrene lid for future measurement of the internal temperature of the hydrating material.

Before mixing the individual components of the measured material, all instruments were switched on, and their functionality was verified. The early switching on of the individual instruments before the actual measurement was also carried out because of current peaks during the electronics start-up, which could subsequently affect some of the measured parameters. The parameters of the generated signal were then set on the Agilent 33220A generator, but the signal generation was not yet activated at this stage. A description of the controlled power amplifier was also set. The computer on the receiving side of the measurement chain was also prepared, with the maximum possible gain set at the beginning of the measurement. The gain was set just above the noise level. This maximum amplification allows the signals to be captured from the beginning to the mould filling.

After the mould and all the electronic equipment were set up, a specific amount of the material under investigation was mixed. After mixing, the mould was filled to the edge. Subsequently, the mould was covered with a polystyrene lid, and the temperature sensor was placed so that it was not in the direct path of the mechanical waves, i.e., so that it was out of the axis of the piezoelectric sensors. The cover was then weighted so that it would not tend to change its position, especially at the beginning of the measurement, and the actual measurement was started.

During the pass-through pulsed ultrasonic method measurements, the signal amplification on the receiving side was carried out manually (we are currently trying to develop automatic software monitoring). The signal was split from the receiving piezoelectric transducer into three channels of the recording device. These channels had different gain settings that were graduated in 10 dB increments. This gradation is essential because the structure of the material changes during hydration, and, thus, the amount of energy that passes through the material. In the case of measurements where manual control is not possible (e.g., overnight), a 20 dB margin is available, which proved to be sufficient in the preliminary experiments. In the evaluation, the three signals are combined into one final signal. Weak or over-amplified parts of the signal were removed ex post.

### 2.5. Effect of Piezoelectric Sensor Protection on Mechanical Waves

Before the actual testing of the selected materials, various calibration tests were performed, and the effect of the created piezoelectric sensor protection on the recorded signals was tested. The piezoelectric sensor protection test looked at the impact that the de-moulding film used does or does not have on the measurements. In the case of placing the probes on top of each other without any protection, the recorded amplitude values came out to be 143 mV, which is lower than when testing the piezoelectric sensors with de-moulding agent and wrapped with foil (see Figure 3). This is because the skinny layer of de-moulding agent creates a perfect acoustic coupling, with the petroleum jelly smoothing out the slight irregularities of the sensor surface. If the transducer and the exciter are placed directly on each other, slight air pockets are created between them due to the unevenness of the two surfaces, and, hence, the voltage obtained is lower. Thus, a higher energy transfer from the exciter to the receiving sensor is achieved. During the pilot measurements, it was found that, for example, it is not the magnitude of the signal amplitude value that is important for determining the possible onset of solidification of a composite material but the time when the signal amplitude reaches its maximum value.

After the measurement setup was built, tested, and calibrated, the second phase of this study began with the actual measurement of the fine-grained cement-based composites.

### 2.6. Measurement Process

In the second phase of the experimental study, the selection and identification of feedstock were carried out. The formulations were designed considering the water coefficient value and the type of cement used. In typical concrete practice, the choice of the kind of cement then depends, for example, on the ambient temperature during the production and processing of the cement composite, the purpose of the structure, the required rate of setting and hardening, the development of the hydration heat, and the exposure environment that will affect the structure or the element itself [25]. Measurements were carried out in which the amplitude changes were monitored, and comparative temperature measurements in the composites were carried out simultaneously with the mechanical wave measurements. The ambient temperature was also measured because the hydration of the cement depends on this temperature, and the cement composite’s maturation rate is also affected by this temperature. The temperature measurements were performed on a recording panel MS6D from Comet System, s.r.o. (Rožnov pod Radhoštěm, Czech Republic) [34,35]. Recording panels are designed for measuring, recording, evaluating, and subsequent processing of input electrical quantities subject to relatively slow changes. In conjunction with appropriate sensors and transducers, they are suitable for monitoring physical quantities (temperature, humidity, pressure, CO_2_, voltage, current, and others).

The measurements were conducted in an air-conditioned room with a temperature of 22 °C and a relative humidity of 45%. The time from the start of the measurement was recorded for all measured variables. Based on the time data from each data logger, all data were then related to a single measurement start, which was most often 600 s from mixing.

Once sufficient data were obtained, the overall data processing was performed using MATLAB programming language scripts, followed by a combination of Microsoft Excel and EasyPlot 4.0.1 for subsequent graphical display.

## 3. Results and Discussion

The results in this text are restricted to the two extreme cases (w/c = 0.33 and 0.60) and water. These results clearly show the differences between mixtures. Additionally, the functionality of the constructed measurement chain and the applicability of the method can be observed. In the summary, the results for all measurements are shown.

### 3.1. Water Alone

Figure 4 and Figure 5 show the results of the measurements of ordinary tap water alone. The graphs show that no significant changes occurred throughout 72 h. The amplitude stabilised around 1.29 V during the measurements, and the difference in water temperature relative to the ambient temperature stayed close to zero. This first experiment verified the stability of the amplitude and temperature values over time on the water alone.

The situation is a little different in the frequency analysis results. There are dominant frequencies in three significant packages (Figure 5 and detailed on Figure 6). Five dominant frequencies were observed, with each colour representing one of the five dominant frequencies. During the measurement, a signal was recorded every 10 s, and, from this, the five dominant frequencies were determined using FFT and plotted on a graph. From these points, significant frequency lines emerged in the graph, producing packages of significant frequencies over time. The first considerable package, especially at the beginning of the measurement, is the frequencies near 850 kHz (the resonant frequencies of the transducer and exciter). The second more significant package is frequencies around 1.5 MHz, which occur throughout the measurement period and are probably related to the water. The third package, associated with the first package, is frequencies around the value of 2 MHz. The shift of the frequencies from 850 kHz to approximately 2 MHz is probably related to the combination of the Doppler effect and interference. Neither the transmitter nor the receiver moves mutually, but, owing to wave folding, the waves resulting from interference and reflections meet after approximately 36 h of measurements. Thus, new wave sources are created in the vessel with the water itself, which affects the frequency of the received waves; thus, the frequency shifts from 850 kHz to 2 MHz.

Figure 6 shows a detailed record of the frequency lines for water alone. Each point shows the dominant frequency in a vertical line for a given time instant. Thus, we recorded five points at each time instant. The individual colours always correspond to the magnitude of the current amplitude at that frequency. For this reason, we have referred to the frequency lines in the comments.

### 3.2. Cement Paste with w/c = 0.33

The cement paste with the lowest water coefficient has a water coefficient of 0.33. Cement paste with this water coefficient is more difficult to mix without adding other additives or admixtures. However, the first graph in Figure 7 shows the dependence of the temperature difference (temperature inside the material—temperature in the laboratory environment) and the amplitude on time. The plot shows a fairly steep increase in amplitude over the time range of 12–40 h. This increase is due to the forming of bonds in the material structure that better transmit the generated waves to the specimen surface. The steady-state amplitude value occurs 72 h after mixing. Once more, the not-very-good mixing influences the internal temperature values in the hydrating specimen very well. Individual peaks are seen as the unmixed cement clusters gradually hydrate. The temperature inside the specimen stabilises at ambient temperature 24 h after mixing.

The dominant frequencies are more sensitive to changes in the structure than the amplitude itself. This is mainly due to the formation of bonds in the material structure. As the cement grains progressively hydrate, different vibrations and nodes are formed, and, therefore, the dominant frequencies, or their positions, are more sensitive than the amplitude of the mechanical waves passing through the structure under study. In the case of a water coefficient of 0.33, settling occurs after approximately 100 h of mixing when frequencies around 1250, 1750, and 2400 kHz are recorded.

### 3.3. Cement Paste with w/c = 0.60

The cement paste with the highest water coefficient (w/c = 0.60) in the baseline measurements behaved differently from the other measurements. The amplitude of the captured signal, as seen in Figure 8, did not stabilise during the observation period, as was typical in the prior cement pastes with lower water coefficients. This behaviour can be explained by the enormously torn structure of this specimen, where the signal did not pass cleanly through the entire structure. The temperature difference stabilised again after about 36 h.

The evolution of the dominant frequencies is also very chaotic, with no obvious frequency lines. A shift is also visible, and this is at 24 h from the mixing of the first relevant frequency records. This behaviour can be explained by forming the first bonds in the cement paste structure, which can transfer the waves from the exciter to the sensor.

### 3.4. Results Summary

Table 1 quantifies the values of the guidelines; i.e., the slope of the amplitude increases from the beginning of the amplitude increase to its initial steady state. A higher value of the directive indicates a steeper growth of this amplitude of the passing signal, indicating a faster formation of the internal structure of the cement pastes and, therefore, an earlier hardening. The deviation from linearity suggests the degree of deviation of the measured signal from the straight line interleaved by this amplitude increase. A higher value indicates a more ‘skeletal’ amplitude increase (i.e., a less linear amplitude increase). The table shows that most of the water coefficients’ maximum deviation was very similar. The cement paste with a water coefficient of 0.45, i.e., the paste closest to the ideal straight line, was the one that could mix the best. In contrast, the cement paste with a water coefficient of 0.60 achieved the worst deviation of the measured signal from the straight line, followed by the cement paste with a water coefficient of 0.40.

In Figure 9, the slope values are plotted against the water coefficient. The differences between the different cement pastes are more prominent. The bar graph shows two groups of cement pastes: one group consists of cement pastes with water coefficients of 0.33, 0.40, and 0.45, and the other group consists of cement pastes with water coefficients of 0.50, 0.55, and 0.60.

For a further comparison of the results, the relative frequency of the dominant frequencies for each cement paste were selected. For greater clarity, the plots were always produced for two cement pastes with similar water coefficient values. The axes of all graphs are identical. The horizontal axis shows the recorded frequencies in MHz, and the vertical axis shows the relative frequency in percentage. At the same time, the resonant frequency of the piezoelectric elements is shown for all graphs, namely, 850 kHz.

The first pair compared in Figure 10 shows subtle differences in the observed dominant frequencies, with only two significant dominant frequencies for the cement paste with a water coefficient of 0.33 and three significant dominant frequencies for the mixture with a water coefficient of 0.40. These significant dominant frequencies correspond to the previous chapters’ main frequency lines. The following graph in Figure 11 shows that, even for a water coefficient of 0.45, these significant dominant frequencies are more numerous than for a cement paste with a water coefficient of 0.50.

In the case of the last comparison graph of Figure 12, this trend of declining significant dominant frequencies is evident. There is only one significant dominant frequency for both cement pastes compared. It can also be seen from these plots that, when the curve has a sharp peak, this dominant frequency is identifiable (e.g., 2 MHz in the case of the cement paste with a water coefficient of 0.45 in Figure 11). In contrast, in Figure 12, the very round peak for the cement paste with a water coefficient of 0.60 is around 1.7 MHz. This “roundness” or “spikiness” in the relative frequency dominance indicates the quality of bonding in the internal structure of the hydrated material. Thus, in the case of a better-quality structure, there are clearly defined dominant frequencies at which the structure oscillates during hydration. In the case of poor-quality internal bonds, it is then impossible to determine at which frequency the material’s internal structure is vibrating. The poor-quality structure is probably formed by a larger number of pores that break the passing waves; thus, it is not possible to clearly measure or determine the significant frequencies of the resulting structure.

### 3.5. Possible Future Developments and Contribution to the Discipline and Practice

In this experimental study, a laboratory setup for ultrasonic pulse measurement was built and tested. This setup allows us to determine the change in amplitude of the mechanical waves passing through a solidifying fine-grained composite (cement pastes of different water coefficients). At the same time, the dominant frequencies of the passing mechanical waves were determined using the fast Fourier transform and were related to the time evolution of the hydration of the cement pastes. Further experiments and research on other types of cement-based or alkali-activated composites will be necessary for this method’s future development and application. For subsequent use, especially in laboratory practice, this non-invasive method can be used as a basic method to determine the degree of hydration of composites. In further analyses and evaluations, it would be helpful to focus on changes in the individual signals recorded due to the composition of the individual mixtures, e.g., changes in aggregate size. It would also be appropriate to apply this procedure to other building materials, e.g., alkali-activated composites, gypsum, etc.

However, it is necessary to reconfigure the experimental setup to enable pilot measurements on coarse-grained composites. This reconfiguration will entail, in particular, enlarging the container for the placement of the fresh mix, where the Vicat ring used in the study will not be suitable in size for the dimensions of the aggregate used. It will probably also be necessary to modify the piezoelectric element to make it usable for higher generated powers. The modification will only be appropriate once it has been determined which existing piezoelectric aspects are suitable for replacement.

The results of this study should do the following:-Expand the capabilities and awareness of acoustic non-destructive methods;-Support efforts to expand the use of measurement experimental methods in the field of non-destructive testing in the construction industry during the early solidification phase of composite materials;-Expand the ability to use existing instrumentation to non-invasively determine the early setting time of composite materials using a simple instrumentation setup.

The pilot experiments performed on a basic set show that the pass-through pulsed ultrasonic method, based mainly on monitoring the evolution of the signal amplitude throughout the measurement, can contribute significantly to a more detailed understanding of the behaviour of composite materials, in particular, to achieve a deeper understanding of the individual processes occurring in the early solidification phase of composite materials and their influence on the mechanical waves passing through. To gain a comprehensive overview of the changes in the structure of the solidifying mixture, we complemented the solution with a frequency analysis of the recorded mechanical waves. This analysis may be particularly useful for relatively quickly and inexpensively determining the initial properties of newly developed mixtures in basic research.

The findings from this study could further develop the possibility of future testing on other composite materials that could not be verified in this experimental study, whether because of time or other reasons. This study should form the basis for further follow-up work aimed at a more in-depth interpretation of the results or at extending the materials tested that were not the focus of this study.

Furthermore, it can be assumed that this method or its modification can provide new insights into the early solidification behaviour of fine-grained cement-based composite materials. This method could also be applied to some other disciplines not primarily related to civil engineering. The possibility of using the pass-through pulsed ultrasonic method, for example, in testing the plasticity of soils, is proving attractive.

## 4. Conclusions

The need to develop new non-destructive methods and procedures [35,36] for testing new materials, whether cement-based [37,38] or alternative binders [39], is a never-ending process. This experimental study applies the pass-through pulsed ultrasonic method during the setting and hardening of fine-grained cement composites. It has been verified that the processes that take place inside cement composites during their hydration affect not only the change in internal temperature, which is related to the evolution of the hydration heat but also to the change in the amplitude and frequency of the mechanical waves that pass through the solidifying material. These changes are related to the internal elasticity of the bonds formed. There is a temporal relationship between the evolution of the internal temperature and the evolution of the amplitude and frequency of mechanical waves during the early stage of the setting of cement pastes. The pass-through pulsed ultrasonic method appears to be a suitable complement or possible substitute for monitoring the solidification and hardening process by the Vicat apparatus in the early stage of building composites, especially in the laboratory environment.

The main advantage of the pass-through pulsed ultrasonic method is its non-destructive and non-invasive nature. The measurements can be repeatable, and treating the fresh mixture is unnecessary before preparing the test specimen. Another advantage of this method is its use for measurements of cement pastes and mortars, and after minor modifications of coarse-grained concrete, which may allow a more profound knowledge of the relationship between the behaviour of the binder itself and the whole composite. The measurement setup can also be used to measure hardening materials that are not cement-based.

A functional setup for applying the pass-through pulsed ultrasonic method to measure these materials was built and tested to analyse the behaviour of mechanical waves passing through fine-grained cement composites during the early hydration phase.

During the construction of the measurement setup, the following selected major complications were addressed:-The protection of piezoelectric sensors from high-humidity mixtures was solved by sufficient insulation with foil and sealing of the specimen mould;-The effect of additional humidity protection on the generated and received signal (if the sensors are not directly attached to the surface, undesired reflections may occur at the transitions between the layers) was solved by applying a bonding agent between the layers and removing air bubbles that were created when the foil was applied to the sensing surfaces of the piezoelectric elements;-Decreasing the signal gain on the receiving side during the solidification process is a complication which arises when a solid internal structure gradually builds up, causing the amplitude to increase to the point where it exceeds the range of maximum values that the recording equipment can detect, and this is resolved by good planning of mixing times and the subsequent monitoring of this level;-Having sufficient energy of the excitation pulse to allow the mechanical waves to pass through the water itself is solved by a gradual change in equipment and testing until the desired result was achieved when the correct configuration was found so that the water itself was measurable.

## Figures and Tables

**Figure 1 materials-17-05971-f001:**
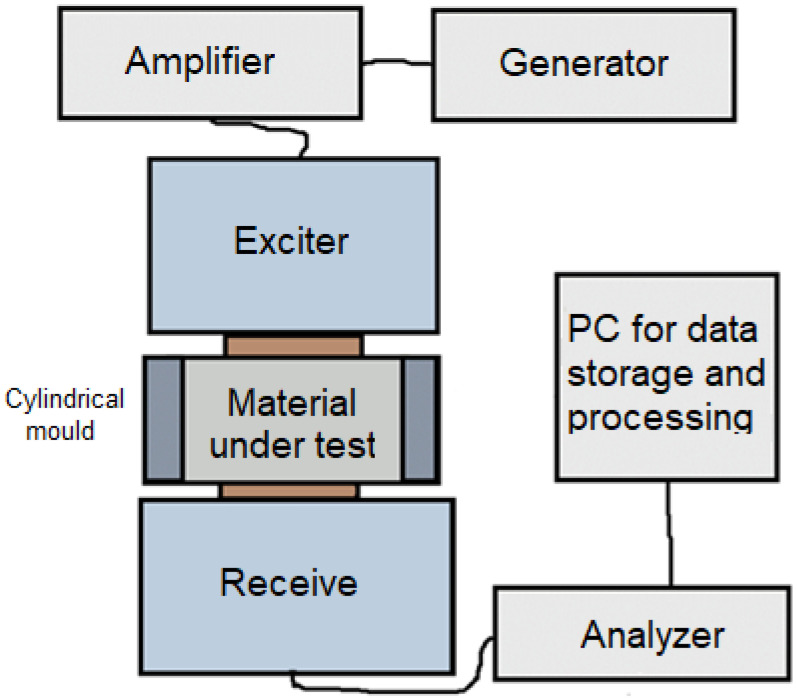
Schematic diagram of pass-through pulsed ultrasonic method for mechanical wave analysis.

**Figure 2 materials-17-05971-f002:**
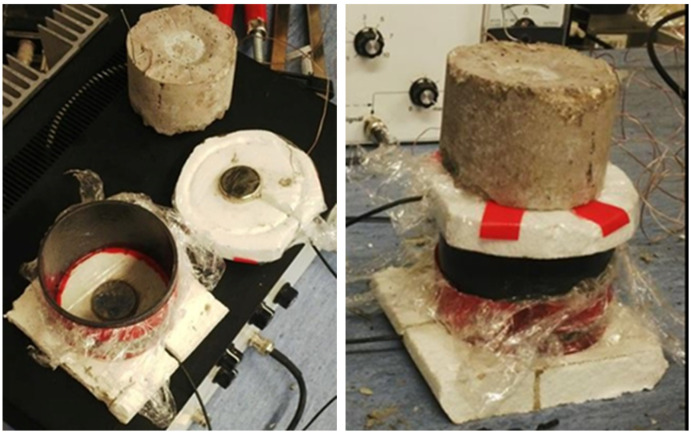
Plastic mould and polystyrene plates with piezoelectric sensors and during measurement (author).

**Figure 3 materials-17-05971-f003:**
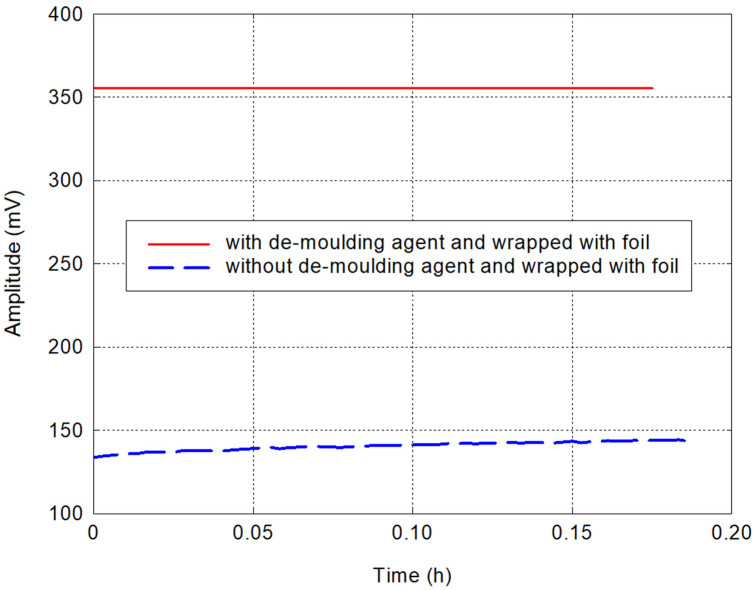
Comparison of recorded amplitude values with/without de-moulding agent and wrapped with foil.

**Figure 4 materials-17-05971-f004:**
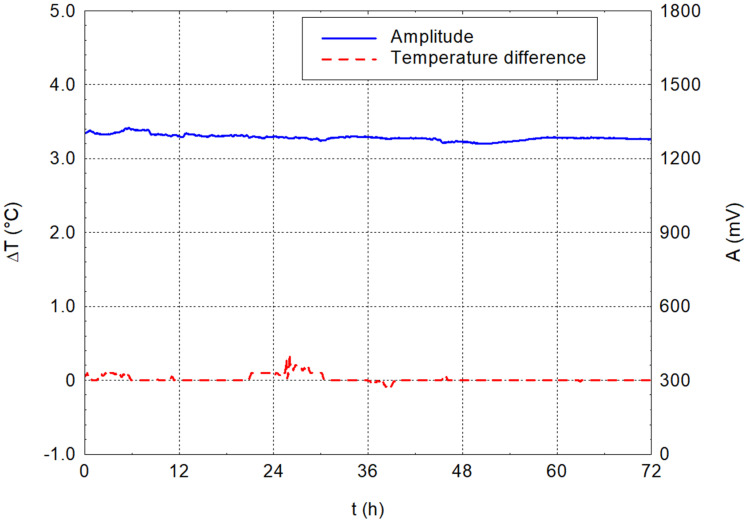
Dependencies of temperature difference and amplitude on time for water.

**Figure 5 materials-17-05971-f005:**
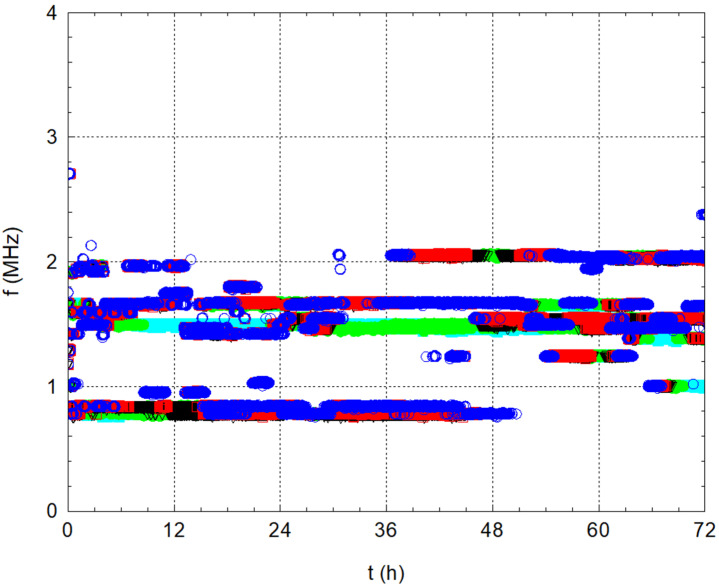
Development of dominant frequencies in time for water.

**Figure 6 materials-17-05971-f006:**
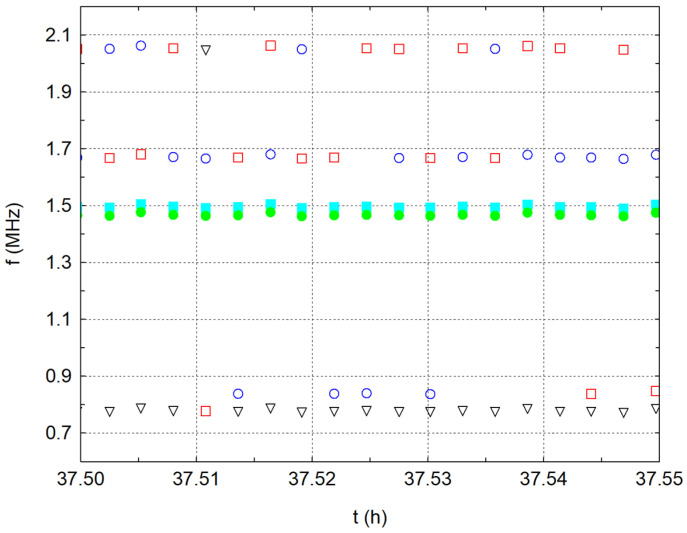
Development of dominant frequencies in time for water—detailed view.

**Figure 7 materials-17-05971-f007:**
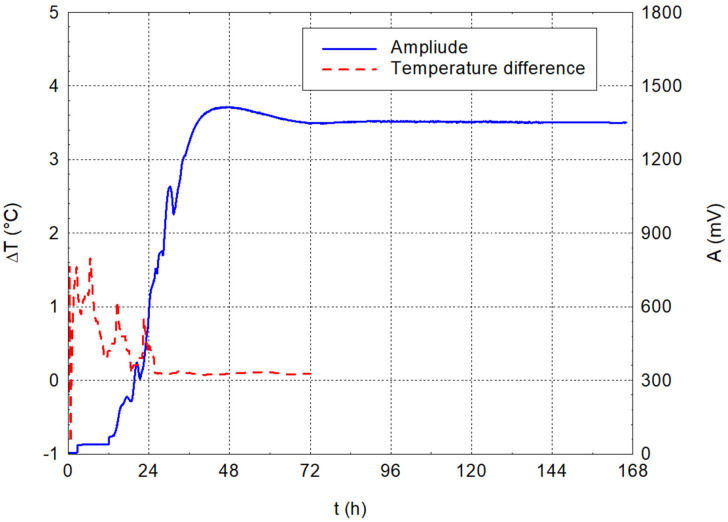
Dependencies of temperature difference and amplitude on time for cement paste with w/c = 0.33.

**Figure 8 materials-17-05971-f008:**
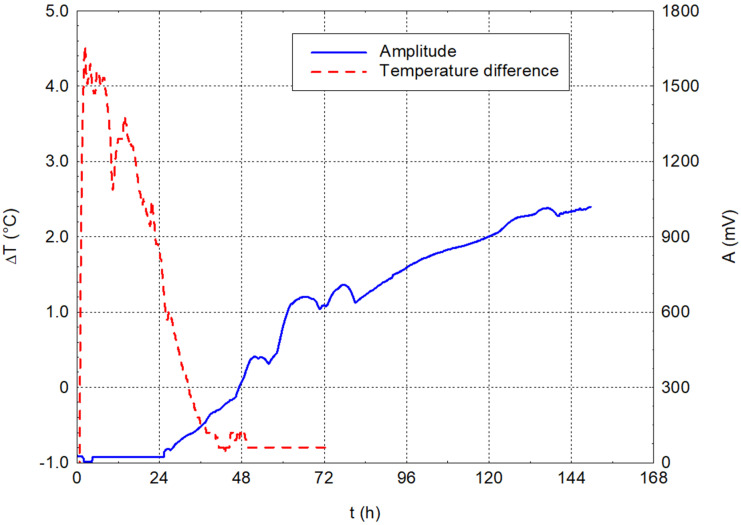
Dependencies of temperature difference and amplitude on time for cement paste with w/c = 0.60.

**Figure 9 materials-17-05971-f009:**
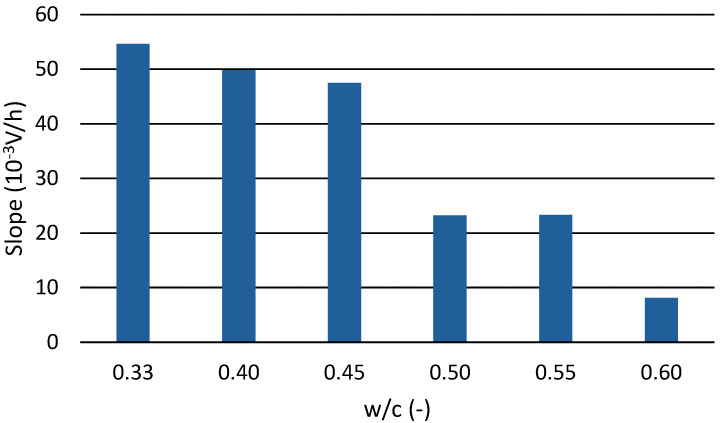
Dependence of the slope of the amplitude on the water–cement ratio.

**Figure 10 materials-17-05971-f010:**
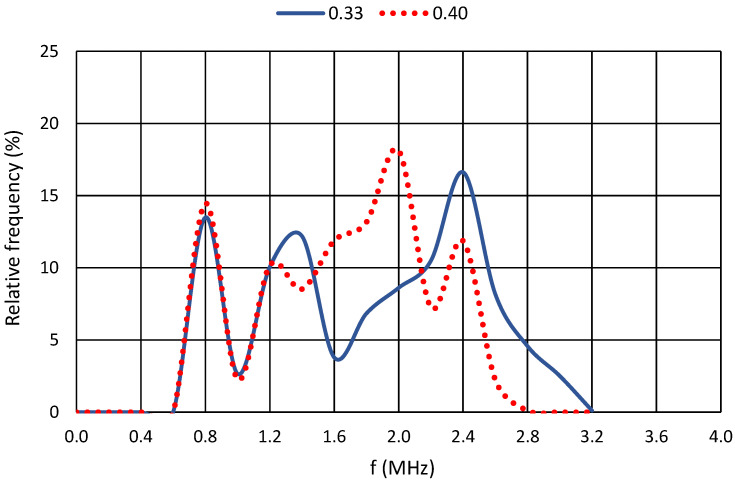
Relative frequencies for cement pastes with w/c = 0.33 and w/c = 0.40.

**Figure 11 materials-17-05971-f011:**
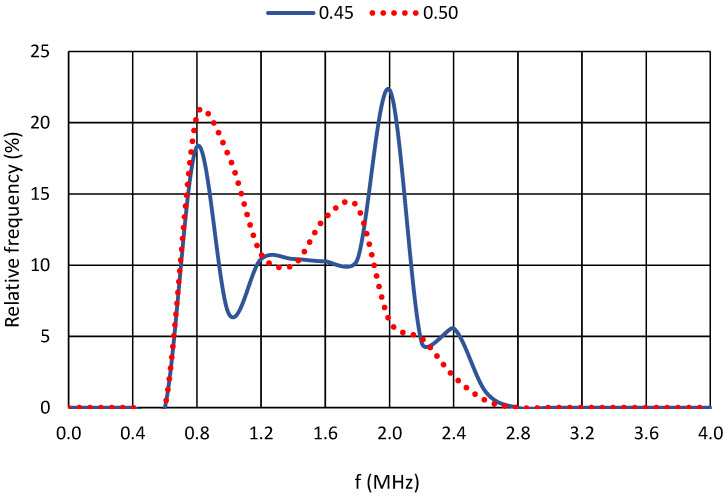
Relative frequencies for cement pastes with w/c = 0.45 and w/c = 0.50.

**Figure 12 materials-17-05971-f012:**
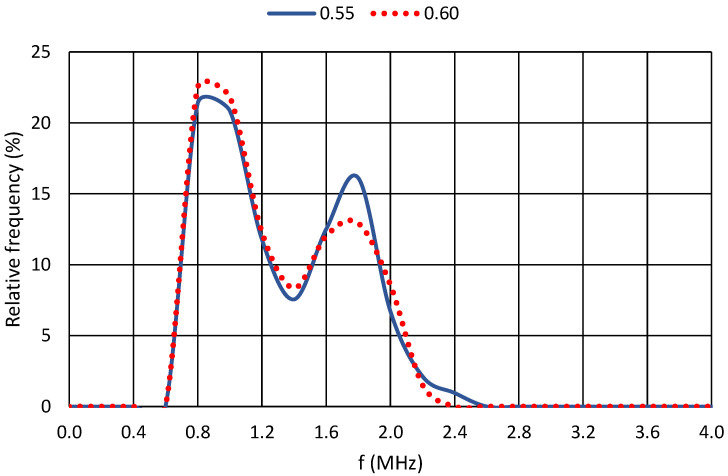
Relative frequencies for cement pastes with w/c = 0.55 and w/c = 0.60.

**Table 1 materials-17-05971-t001:** The amplitudes’ slope and their deviation from linearity.

w/c	Slope (Time Interval for Determining the Directive)	Maximum Deviation from Linearity
-	10–3 V/h	-
0.33	54.6 (12–40 h)	160
0.40	49.8 (12–43 h)	168
0.45	47.5 (10–37 h)	87.6
0.50	25.2 (12–63 h)	163
0.55	23.3 (13–60 h)	165
0.60	8.1 (24–144 h)	170

## Data Availability

The original data presented in this study are included in the article. Further inquiries can be directed to the co-author Libor Topolář by email.

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
