# Peer review of "Experimental Study of Mechanical Wave Propagation in Solidifying Cement-Based Composites"

_materials, 2024, doi:10.3390/ma17235971_

Round 1

Reviewer 1 Report

Comments and Suggestions for Authors

Authors studied on the propagation of mechanical waves during the solidification process of cement-based composites. It employs a pass-through pulsed ultrasonic method to monitor the setting and hardening stages of fine-grained cement composites. The study uses piezoelectric transducers to capture the mechanical waves passing through the material during solidification. Over time, the amplitude and frequency changes of these waves are analyzed, providing deeper insight into the structural development of the material. The focus of the study is on cement pastes with different water-cement ratios (w/c = 0.33~0.60). The results show significant differences in wave propagation behavior, corresponding to the solidification process of each mixture. This ultrasonic method shows potential for evaluating the hydration and hardening processes of cementitious materials, with possible applications in non-destructive testing during the early stages of construction. The findings support further exploration of the method’s applicability to other composite materials. However, the current form of this study cannot be acceptable. Some aspects as listed below:

1.      In the part of 2.2 ultrasound method, it is suggested to add notes to the actual test specimen picture in Fig.3 to explain the specific functions of each part of the specimen and the corresponding relationship with each part in Fig.2.

2.      The study focuses on cement pastes with water-cement ratios between 0.33 and 0.60. However, the rationale for selecting this range is not clearly stated. Is this range sufficient to cover the typical conditions encountered in practical applications? Should the range be expanded to better represent different operational scenarios?

3.      Furthermore, the paper does not mention the number of experimental repetitions or include statistical analysis of the results, particularly in terms of the consistency of findings under different water-cement ratio conditions.

4.      Under certain water-cement ratio conditions, significant frequency fluctuations were observed, and clear dominant frequency lines were not formed, especially at higher water-cement ratios. It is recommended to provide a more detailed analysis of this frequency instability, considering whether it may be related to macro phenomena such as paste separation or increased porosity.

5.      The paper mentions the use of protective films and release agents to isolate the transducers from the material. Please clarify whether such treatments may have affected the transmission and reception of mechanical waves.

6.      While the paper uses FFT analysis to track changes in dominant frequencies over time, the explanation of how these frequency changes relate to the microstructural evolution during cement hydration is insufficient. It would be beneficial to include more discussion on how these frequency changes reflect internal structural developments, particularly in relation to the formation of hydration products (e.g., C-S-H gel and portlandite). Consider combining this analysis with XRD or SEM to validate the correlation between frequency changes and structural evolution, thereby enhancing the explanatory power of the method.

7.      The paper records changes in internal temperature during the solidification process but does not explore how these temperature changes affect wave propagation characteristics. Temperature can influence not only the rate of hydration reactions but also the material’s elastic modulus and wave velocity. It is suggested to further quantify the effect of temperature on wave velocity, attenuation, and other parameters through experimental or simulation approaches to strengthen the study’s conclusions.

Author Response

Thank you for your comments and recommendations. Everything is in the attached Word file. Sincerely, authors' teams.

Reviewer 2 Report

Comments and Suggestions for Authors

Materials paper:

Experimental Study of Mechanical Wave Propagation in Solidifying Cement-Based Composites

By Jakubka et al (2024)

Opinion’s reviewer:

The document presented should be written following the guidelines of the journal (see section and subsection headings, see line 230; references see line 223 it is [17…

But more, this paper should be rewritten in 2 papers: one for the novel development of measurement method with an example of application (one sample of composite) and another one tested samples and explanations on the results obtained regarding the benefits of this method compared to others (literature). The second could be proposed but authors could shorten the paper to around 15pages.

Detailed comments:

Title: ok it is clearly mentioned that is an experimental study (using a novel monitoring?)

Abstract:

In the abstract, it is clearly indicated that is an experimental study (repeated) using “a measurement system using the pass-through pulsed ultrasonic method for fine-grained cement composites built and tested during the early stages of the setting”.

The abstract is not well written: what are the novelties: measurement system and/or observations at early age of the hydration process; what type of composites tested; what are the main results and conclusions. The abstract must be rewritten.

Introduction:

The authors should if possible to use too many brackets.

The introduction is not well prepared: too many sentences are not useful such lines 34-38.

Introduction must include literature on the topic of the paper (literature must be completed on this aspect): on the use of different measurement methods for observations on the early-aged composites (are they cementitious materials or others?) and the development of a novel monitoring. It is interesting to give advantages-disadvantages of each method used. And after the introduction should focus on the novel method and composites tested (methodology).

The problem: what is the most important for authors 1-the development of the novel method (if yes, the journal materials is only appropriate if they add one test for explaining the results on a (only one) composite) 2- the results obtained on different composites with this method (in this case novelty is more in the observations on hydration phenomenon).

Theoretical background:

Too long, in this paper, the based equation should only recall in a section as “development a novel method” with the scheme of the final monitoring and procedure used.

For example what is the usefulness of figure 3?

The subsections : “Characteristics of the investigated materials” and “Hydration (setting and hardening) of cement” could really be shortened: all information are well-known.

A subsection as “Experimental background” is not well appropriate here. A subsection on the methodology (monitoring, procedure) used with a list of samples of materials tested is more indicated.

(Details of the review are stopped here).

Author Response

(The authors gave the same response as above.)

Reviewer 3 Report

Comments and Suggestions for Authors

This study deals with an experimental application of the pass-through pulsed ultrasonic method to fine-grained cement composites during solidification and hardening. A specific measurement system was developed and tested on samples having different water-cement ratios, with the aim of expanding the use of acoustic methods as non-destructive testing (NDT) and supporting the construction industry in the field of composite materials.

Generally, the paper results well structured, comprehensive in its information and very interesting. 

Point 1. To date there are several works addressed to estimate material properties by means of sonic and ultra sonic tests. Among the others, Authors may consider the following works:

doi.org/10.1080/15583058.2021.2015641

doi.org/10.1007/s10921-020-00693-2

Point 2. In the introduction the novelty of the paper should be better stressed, considering that the investigation proposed may be a valid alternative with respect to the destructive tests. 

Point 3. Reviewer does not understand how  the knowledge of the ultrasonic pulse measurement may be applied. This aspect is not clear in the paper. Please comment on this.

Point 4. Throughout the paper is paid attention to the influence of the w/c content. In practical application, how it should be estimated, in order to correlate this parameter to the ultrasonic pulse velocity? Moreover, does it exist in somehow a correlation with some material mechanical property? Please, comment on this.

Point 5. Check the abstract. Please, review sentence in line 17-18. 

Point 6. The text contains some repeated words and/or sentences (for instance see line 36, lines 210-211 compared with lines 216-217).

Point 7. It is suggested to write the sentence on lines 77-82 in an easily readable manner. Furthermore, check sentence on lines 300-301. It seems incomplete.

Point 8. If possible, include in the literature some scientific references for the statement on lines 535-537, if any. 

Point 9. If possible, please increase image resolution, (Figure 1, Figure 3, Figure 9, Figure 11, Figure 12, Figure 13 and Figure 15).

Point 10. Conclusion should be revised in order to highlight only the main outcomes of the paper. 

Author Response

(The authors gave the same response as above.)

Round 2

Reviewer 1 Report

Comments and Suggestions for Authors

Authors have revised the manuscript according to some comments. However, the current form of this study cannot be acceptable. Some aspects as listed below:

1.        Due to potential sample variability in the hydration process of cement-based materials, it is recommended to include more experimental repetitions. The paper currently presents a limited analysis of experimental repeatability, mentioning only preliminary manual gain adjustments and signal monitoring.

2.        Although the paper employs Fast Fourier Transform (FFT) for spectral analysis, it primarily focuses on observing the general trend of frequency changes. It is suggested to incorporate additional spectral analysis metrics, such as signal attenuation coefficient and phase variation, to provide deeper insights into the relationship between internal structural changes and wave propagation characteristics.

3.        The study predominantly uses FFT to analyze frequency changes; however, FFT has limitations in handling non-stationary signals, especially during the early hydration phase of cement-based materials, where the signals often exhibit non-linear and time-varying characteristics. It is recommended to introduce more advanced time-frequency analysis methods, such as Wavelet Transform or Short-Time Fourier Transform (STFT), to achieve better resolution in both time and frequency domains, thereby capturing dynamic changes during the early hydration process more accurately.

4.        Significant differences in frequency change trends were observed for mechanical wave signals under water-cement ratios of 0.33 and 0.60. It is recommended to further explain these results by integrating the cement hydration heat model and changes in pore structure. Additionally, combining the analysis with existing hydration kinetics models could provide a more comprehensive theoretical framework for understanding the causes of characteristic frequency changes and enhance the theoretical support for the findings.

5.        During the analysis of water sample experiments, the authors mention the concept of "water memory" and cite a non-scientific source. This part lacks scientific basis and should be removed. Instead, a scientifically sound explanation should be provided.

6.        The paper notes the use of protective films and coupling agents (e.g., petroleum jelly) to prevent moisture effects on the transducers. This treatment could introduce additional interface reflections, particularly for high-frequency ultrasonic signals, as the film layer may result in inconsistent acoustic coupling and affect signal stability and measurement accuracy. It is suggested that future studies employ more consistent and reliable coupling agents, such as silicone oil or specialized acoustic couplants, and conduct comparative tests to correct for errors introduced by transducer coupling.

Author Response

Thank you for your review. We appreciate your comments, but, unfortunately, not everything can be changed or accepted. The reasons are lack of time, inability to repeat some measurements due to a change in the configuration of the measurement setup for the ongoing project, and different batches of cement (change in microstructure). Furthermore, the paper is only intended to show a possible technical solution and, as such, does not currently aim to understand the effect of the change in structure on mechanical waves in detail. We hope that you are satisfied with our responses to your suggestions.

Reviewer 2 Report

Comments and Suggestions for Authors

Materials paper:

Experimental Study of Mechanical Wave Propagation in Solidifying Cement-Based Composites

By Jakubka et al (2024)

Opinion’s reviewer:

Authors have shortened the paper only by suppressing some paragraphs. So the document has not been rewritten and not improved. The text is difficult to read because many redundant sentences, confuse sections or lack of information (on materials tested).

This paper is not publishable and it will be now difficult to get a correct scientific paper.

Detailed comments:

Title: As the paper has been shortened, a new title more representative of the paper should be given, with a focus on the innovative monitoring.

Experimental study is not only the adequate term. Title should begin as “A new or an innovative method for….”.

Abstract:

Line 12: it is totally unclear: what are really the objective demonstrating innovation? New measurement technique or analysis of measurements? This line should be suppressed because the second sentence have the same signification.

The abstract is not well written, lacks of clarity.

Introduction:

Some brackets are not useful.

Line 37: more and more, cement is abandoned (not so eco-friendly) and low-carbon binders and geopolymers are investigated.

Lines 81-82: not in introduction; the perspective of application should be given after conclusions or with conclusions for future research works.

Introduction is not well clearly written: literature on the similar technique-methodology used for the novel technique (explaining the innovative part)-tests of application-conclusions.

Methods and materials:

Lines 84-130: these paragraphs concern literature on the measurement technique; these lines must be inserted in literature in introduction; in this section the novel technique must only described and presented.

Pass-through pulsed ultrasonic method:

This subsection was not numbered.

Materials:

This subsection was not numbered..

Lines 186-195: what is a natural mortar? These paragraph is not useful. It is well known for each civil engineer and all civil engineers know the basic mixes as paste (or sometimes called cement paste); mortar and concrete.

Lines 197-198 and 203-204: it is redundant sentences!

This section is “materials”; so the authors should explain what kind of materials they tested? In this section they explain what factors influence the properties of concrete: it is well-known and it is possible to cite only some reference books such the Adam Neville, J.P. Ollivier et al; ….). They should report here in this section the material(s) tested only.

Measuring chain:

This subsection was not numbered..

This subsection and the 2 following ones, are linked to the “development of the measurement technique” (novel technique) and these subsections belong to the section (method, or measurement technique or something equivalent). The authors should separate: measurement technique and material tested.

Line 230: the authors wrote “fine grained composites (not clearly defined). Concrete is not a fine-grained material and “fine” needs to be clearly defined by the limit size of aggregates (sand if it is mortar).

Lines 254-255: unclear: fine-grained composite is not only cement-based: bio-based materials are also fine-grained composite with vegetal aggregates!

Figure 3: put on figure some indications and dimensions; indicate view (A) and (B) on figures and in legend. If we consider the photo on the right: nothing in materials section was written about the preparation, molding of the mortar or fine-grained material?

Lines 270-277: a clear scheme will be more useful for readers.

Preparation and course of the measurement:

Authors should distinguish “tested material” and “measurement technique” in separate section in material(s) and method(s).

Lines 326-332: this paragraph concerns  preparation of the tested materials”. Till here, we do not know anything about the tested material!

Effect of piezoelectric sensor protection on mechanical waves:

Line 344: we do not know what type of materials is selected?

Measurement process:

Lines 367-372: we do not know what type of materials tested? More we read about concrete and it was written many times in the above text: tested materials are fine-grained materials?

Results and discussion:

All comments are stopped here.

Comments on the Quality of English Language

To be seriously improved

Author Response

Dear reviewer, we appreciate your opinions, but we still feel from them that you are forcing yourself into the review and not being completely objective. Many of your comments stem from inattentive reading and misunderstanding of the text as a whole. We have other things to do than take the time to address your not-always-valid comments. Sincerely, the team of authors

Reviewer 3 Report

Comments and Suggestions for Authors

The paper has been revised according to the comments provided. Therefore, it may be considered for the publication in the current revised form.

Author Response

Thank you for your review and decision.

Round 3

Reviewer 1 Report

Comments and Suggestions for Authors

Authors have revised the manuscript according to comments. The current form of this study can be acceptable. 

Author Response

Thank you for your review and decision.